# Unifying description of competing orders in two-dimensional quantum magnets

Xue-Yang Song[1], Chong Wang[1,2], Ashvin Vishwanath[1] & Yin-Chen He[1,2]

Quantum magnets provide the simplest example of strongly interacting quantum matter, yet they continue to resist a comprehensive understanding above one spatial dimension. We explore a promising framework in two dimensions, the Dirac spin liquid (DSL) — quantum electrodynamics ($QED_3$) with 4 Dirac fermions coupled to photons. Importantly, its excitations include magnetic monopoles that drive confinement. We address previously open key questions — the symmetry actions on monopoles on square, honeycomb, triangular and kagome lattices. The stability of the DSL is enhanced on triangular and kagome lattices compared to bipartite (square and honeycomb) lattices. We obtain the universal signatures of the DSL on triangular and kagome lattices, including those of monopole excitations, as a guide to numerics and experiments on existing materials. Even when unstable, the DSL helps unify and organize the plethora of ordered phases in correlated two-dimensional materials.

[1] Department of Physics, Harvard University, Cambridge, MA 02138, USA. [2] Perimeter Institute for Theoretical Physics, Waterloo, ON N2L 2Y5, Canada. Correspondence and requests for materials should be addressed to A.V. (email: avishwanath@g.harvard.edu)

I n recent years, gauge theories have been increasingly used to describe quantum magnets, particularly when geometric frustration leads to an enhancement of quantum fluctuations[1,2]. In these situations, classical descriptions are usually inadequate and entirely new 'quantum spin liquid' phases can emerge, described by deconfined charges of the gauge theory. Even when ordered states appear, the quantum interference between different orders has no classical analog, but can be captured by a gauge theory[3,4]. Progress in understanding quantum magnets will have ramifications well beyond the insulating state and could explain nearby conducting phases obtained on doping[5,6], including the high-temperature superconductors seen in diverse systems from the copper oxide materials to the recently realized twisted bilayer graphene[7,8].

Previous gauge theory-based approaches have pursued different approaches for bipartite and non-bipartite lattices. For example, starting with a Schwinger boson-based representation of spins, a $Z_2$ gapped spin liquid is the natural 'mother' state for describing non-bipartite lattices[9–13], while the similar procedure for the bipartite case indicated Neel and valence bond crystal phases for bipartite lattices, separated by a deconfined quantum critical point[3,4,14]. These paradigms which are ultimately based on quantum disordering an initial classical ordered state (non-colinear versus colinear order on the non-bipartite and bipartite lattices, respectively) represent significant progress towards a synthesis. However, here we will argue that an essentially quantum parent state, the $U(1)$ Dirac spin liquid (DSL) can capture these insights and can further bridge bipartite and non-bipartite lattices using a common framework.

Although there is a considerable theoretical literature on the $U(1)$ Dirac spin liquid, the properties of a crucial class of excitation, the magnetic monopoles, have not, until now, been systematically explored on different lattices. Here we compute the monopole symmetry quantum numbers, which then allows us to make considerable progress in understanding these remarkable states. The magnetic monopoles are 'instanton' excitations, that occur at points in the $2 + 1$D spacetime. When they can be ignored (i.e. when irrelevant in the RG sense), the Dirac spin liquid can be stabilized, resulting in a significantly enlarged symmetry. In addition to the conformal invariance of the fixed point, a defining feature of the DSL is the emergence of a $\sim U(4)$ symmetry at low energies, which incorporates both spin and lattice symmetries.

A useful analog in one lower dimension is the Luttinger liquid, also a gapless phase that describes quantum liquids in $1 + 1$D. There too, the stability of the phase is threatened by instanton excitations in the form of vortex tunneling events. Symmetry transformation properties of the instanton insertion operators play a key role in determining both stability and the nature of the ordered phases which result following instanton proliferation. The phase diagram of quantum spin chains[1,15] and the superfluid Mott transition of one-dimensional bosons[16] can be understood in these terms. When instantons are irrelevant, the gapless Luttinger liquid is stabilized, but when they proliferate, a gapped phase, such as the valence bond crystal or Mott insulator is obtained. In fact, the Luttinger liquid theory can be reformulated in terms of Dirac fermions coupled to a $U(1)$ gauge field, and hence can be viewed as the one-dimensional version of $U(1)$ Dirac spin liquid[17–19].

Similarly, our computation of monopole quantum numbers sheds light on key issues such as: Can the Dirac spin liquid be a stable ground state and if so what are its key experimental signatures? If unstable, what are the likely alternate phases that are stabilized in its place? What is the underlying difference between bipartite and non-bipartite lattices?

Previously, early work on the square lattice quantum antiferromagnet, inspired by the copper-oxide materials, studied the staggered flux and $\pi$-flux mean field theories[20–23] within the fermionic representations of spins. Renewed interest emerged when analogous states on the kagome lattice were introduced in refs. [24–28]. The effect of fluctuations were studied in several works[29,30] and a dictionary relating fermion bilinears to local operators and the enlarged symmetry of the Dirac spin liquid were emphasized in refs. [28,31]. Recently, the Dirac spin liquid on the triangular lattice has been studied[32–35]. However, most works have ignored the monopole excitations and their symmetry properties, with a few exceptions[28,36–39]. In ref. [40], the conceptual framework to study monopoles with fermion zero modes was introduced. The important role of the Dirac sea Berry phase for spatial symmetries was invoked in ref. [36], while numerical calculations of projected wavefunctions revealed properties of monopoles in refs. [28,37]. A discussion of monopole symmetry properties on the square lattice and a numerical evaluation was reported in ref. [38]. A trivial monopole was found, which appeared to be in conflict with the Lieb–Schultz–Mattis–Oshikawa–Hastings theorem (LSMOH)[41–43]. It was understood more recently that even with a trivial monopole, the QED$_3$ theory with $N_f = 4$ still possesses a symmetry anomaly that forbids a trivial vacuum, in agreement with LSMOH theorem[44].

Here, we calculate symmetry transformation of monopoles paying special attention to the subtle 'Dirac sea' contributions, which arise from the Berry's phase acquired by monopoles on moving around gauge charges of the filled Dirac sea. The lattice provides a short distance cutoff that allows us to calculate this contribution, which proves crucial to the physics. We show that a trivial monopole appears in the square lattice DSL, consistent with duality-based arguments[44] and earlier calculations[38] and does not contradict the LSMOH theorem. We then extend these arguments to other lattices where the arguments and methods of refs. [38,44] do not apply. The DSL on triangular and kagome is more stable against monopole proliferation and proximate ordered phases descend naturally from monopole quantum numbers, e.g., the familiar 120º magnetic orders can be obtained from monopole proliferation. Physical consequences of these calculations for numerical simulations and experiments are then discussed.

## Results

**Gauge theory description of spin systems and monopoles.** We will be interested in spin-1/2 systems on various two-dimensional lattices. Let us briefly review the fermionic spinon decomposition of spin-1/2 operators on the lattice which will lead us to the desired gauge theory description. We first decompose the spin operator

$$\mathbf{S}_i = \frac{1}{2} f_{i,\alpha}^\dagger \sigma_{\alpha\beta} f_{i,\beta}, \qquad (1)$$

where $f_{i,\alpha}$ is a fermion (spinon) on site $i$ with spin $\alpha \in \{\uparrow, \downarrow\}$ and $\sigma$ are Pauli matrices. This re-writing is exact if we implement the constraint $\sum_\alpha f_{i,\alpha}^\dagger f_{i,\alpha} = 1$. To make progress, we consider a mean field approximation that only imposes the constraint on average, followed by a discussion of fluctuations (see ref. [2]),

$$H_{\mathrm{MF}} = -\sum_{ij} t_{ij} f_i^\dagger f_j. \qquad (2)$$

There is a gauge redundancy $f_i \rightarrow e^{i\alpha_i} f$ in the parton decomposition Eq. (1), which results in the emergence of a dynamical $U(1)$ gauge field $a_\mu$ that couples to the fermions $f$, i.e. $t_{ij} \rightarrow t_{ij} e^{ia_{ij}}$. The non-bipartite nature (second-neighbor hopping on bi-partite lattice) is needed to make sure that the gauge group is $U(1)$ rather than $SU(2)$[2]. Next, we arrange the hopping term $t_{ij}$ such that the hopping model leads to a pair of Dirac nodes, per

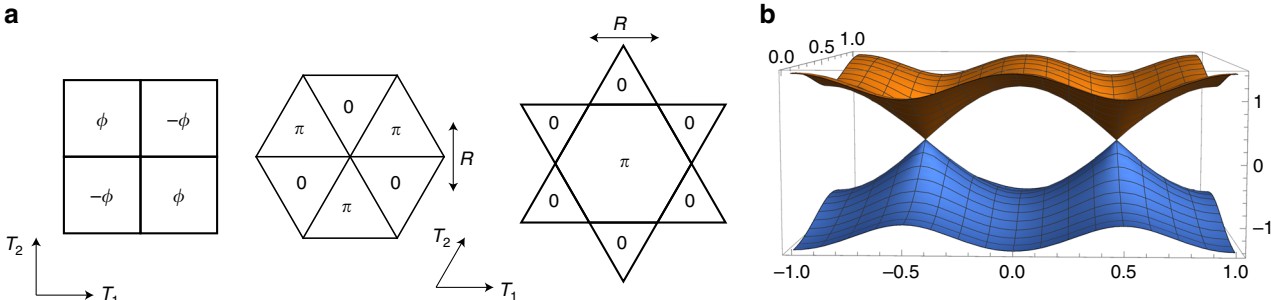

**Fig. 1** Mean-field ansatz and band structure. **a** Mean-field ansatz of Dirac spin liquid on the square, triangular, and kagome lattice. Mean-field Hamiltonian has only nearest hopping with a flux in each plaquette. **b** Band structure of the square lattice $\pi$-flux mean field ansatz with two Dirac cones at the momentum points $(k_1, k_2) = (\pm\pi/2, \pi/2)$

spin, at half filling. Such Dirac dispersions, with four flavors of Dirac fermions (with two spin and two 'valley' labels) can be realized on the honeycomb lattice with only nearest hopping, as well as on other lattices (square, kagome, and triangular lattice) with appropriate choice of $t_{ij}$ as shown in Fig. 1. We note that the mean-field Hamiltonian actually breaks lattice symmetry, but the spin liquid state has all the lattice symmetry after we incorporate the gauge constraint. For example, the triangular lattice ansatz can be $C_6$ invariant if we supplement space group operation with an $SU(2)$ gauge transformation $(f_{i,\uparrow}, f_{i,\downarrow}^\dagger)^T \to i\sigma^2 (f_{i,\uparrow}, f_{i,\downarrow}^\dagger)^T$.

In the low energy, long wavelength or infrared (IR) limit, the theory reduces to the following Lagrangian:

$$\mathcal{L} = \sum_{i=1}^{4} \bar{\psi}_i i \not{D}_a \psi_i, \tag{3}$$

where $\psi_i$ is a two-component Dirac fermion with four flavors labeled by $i$, and $a_\mu$ is a dynamical $U(1)$ gauge field. We choose $(\gamma_0, \gamma_1, \gamma_2) = (i\mu^2, \mu^3, \mu^1)$, where $\mu$ are Pauli matrices. The staggered flux state on square lattice has velocity anisotropy, but it is irrelevant at large $N$ limit[31]. This theory is also known as quantum electrodynamics in three space–time dimensions, $QED_3$ with four fermion flavors $N_f = 4$. The theory as written implicitly assumes that the $U(1)$ gauge flux, i.e. the total flux of the magnetic field, $j_\mu = \frac{1}{2\pi} \varepsilon_{\mu\nu\lambda} \partial_\nu a_\lambda$ is conserved. This theory, sometimes referred to as noncompact $N_f = 4$ $QED_3$, flows to a stable critical fixed point in the IR, as supported by recent numerical studies[45].

However, it is clear that this conservation of flux cannot be the consequence of a microscopic symmetry. Our model has no corresponding $U(1)$ symmetry at the microscopic level, which is an artifact of a topological conservation law in the low-energy model (hence we refer to this as $U(1)_{\text{top}}$). This is remedied by allowing for quantum tunneling between vacuaa of different total flux. The tunneling events, instantons, occur at space–time points and the corresponding operators that create (or destroy) $2\pi$ flux quanta and are termed monopole (anti-monopole) operators. Unlike most other physical operators, the monopole cannot be expressed as a polynomial of fermion or gauge fields. Nevertheless, it is important to note that these operators are local: they modify the magnetic field locally and the inserted $2\pi$ flux is invisible at large distances. As such they should be included in our Lagrangian Eq. (3). A key question will be whether physical symmetries might restrict which monopole operators are allowed. For this, we need to take a more careful look at the monopole operators.

**Monopoles and zero modes.** Previous calculations of monopole quantum numbers in gauge theories of quantum magnets have largely focused on bosonic $QED_3$ where the spinions are bosons, such as in the $CP^1$ models of quantum magnetism[3,46,47]. There

monopoles play a key role in descriptions of the Neel to valence bond solid (VBS) transition in square lattice quantum antiferromagnets. The case of fermionic $QED_3$ is in many ways richer, one of which is the presence of monopoles zero modes. We will see that this allows for a wider description of physical phenomena. For example, both collinear and non-collinear magnetic orders can be captured within a single theory, in contrast gauge theories of bosonic spinons capture one or the other, depending on the nature of the gauge group[3,11,48].

In a theory with massless Dirac fermions, monopoles occur along with fermion zero modes. Recall, massless Dirac fermions in a magnetic field form Landau levels, in particular a zero energy Landau level with a degeneracy equal to the number of flux quanta $\Phi/2\pi$. Thus, addition of $2\pi$ flux creates a fermion zero mode for each Dirac fermion flavor, hence with $N_f = 4$ we expect four zero modes. To maintain neutrality of gauge charge, we must fill half these modes, which can be done in $C_2^4 = 6$ ways. So a monopole operator can be schematically written as

$$\Phi \sim f_i^\dagger f_j^\dagger \mathcal{M}_{\text{bare}}^\dagger, \tag{4}$$

where $f_i^\dagger$ creates a fermion in the zero-mode associated with $\psi_i$, and $\mathcal{M}_{\text{bare}}$ creates a bare flux quanta without filling any zero mode. These operator can be more precisely defined, through state-operator correspondence, as states of the $QED_3$ theory defined on a two-sphere $S^2$ with a $2\pi$ background flux[40]. The Dirac zero-modes have zero angular momenta on the sphere in a $2\pi$ background flux. This implies that the monopoles are scalars under the Lorentz group. In terms of the $SU(4)$ flavor symmetry, they form a vector of $SO(6) = SU(4)/\mathbb{Z}_2$. Note, the monopoles are in contrast to all other gauge invariant operators, such as fermion bilinears only transform under $SU(4)/\mathbb{Z}_4$ without carrying the $U(1)_{\text{top}}$ charge. Thus, all physical operators are invariant under the combined action of the center of SO(6) and a $\pi$-rotation in $U(1)_{\text{top}}$. So the precise global symmetry group is:

$$\frac{SO(6) \times U(1)_{\text{top}}}{\mathbb{Z}_2} \tag{5}$$

together with the Lorentz group and discrete symmetries $\mathcal{C}_0, \mathcal{T}_0, \mathcal{R}_0$ of the $QED_3$ Lagrangian (3). One can certainly consider $2\pi$-monopoles in higher representations of SO(6), but in this work we will assume that the leading monopoles (with lowest scaling dimension) are the ones that form an SO(6) vector—this is physically reasonable and can be justified in large-$N_f$ limit.

Instead of working with the explicit definition of monopoles from Eq. (4), we shall simply think of the monopoles as six operators $\{\Phi_1, ..., \Phi_6\}$ that carries unit charge under $U(1)_{\text{top}}$ and transform as a vector under SO(6): $\Phi_i \to O_{ij}\Phi_j$. Clearly the lattice spin Hamiltonians does not have such a large symmetry—typically we only have spin rotation, lattice symmetries (lattice

**Table 1 Summary of monopole transformations on square (staggered flux state) and honeycomb lattices, where $\Phi_{1/2/3}$ are spin singlet monopoles ($\Phi_\pm = \Phi_1 \mp i\,\Phi_2$) and $\Phi_{4/5/6}$ are spin triplet monopoles**

| Lattice | $T_1$ | $T_2$ | $R_x$ | Rotation | $\mathcal{T}$ | Note |
|---|---|---|---|---|---|---|
| Square | $\Phi_1^\dagger$ | $\Phi_1$ | $-\Phi_1$ | $-\Phi_1$ | $-\Phi_3$ | $\Phi_1^\dagger$ | $Re[\Phi_1]$ as $\overline{\psi}\tau^3\psi$ |
| Square | $\Phi_2^\dagger$ | $-\Phi_2$ | $-\Phi_2$ | $-\Phi_2$ | $-\Phi_2$ | $-\Phi_2^\dagger$ | $Im[\Phi_2]$ trivial |
| Square | $\Phi_3^\dagger$ | $-\Phi_3$ | $\Phi_3$ | $\Phi_3$ | $\Phi_1$ | $\Phi_3^\dagger$ | $Re[\Phi_3]$ as $\overline{\psi}\tau^1\psi$ |
| Square | $\Phi_{4/5/6}^\dagger$ | $-\Phi_{4/5/6}$ | $-\Phi_{4/5/6}$ | $\Phi_{4/5/6}$ | $\Phi_{4/5/6}$ | $-\Phi_{4/5/6}^\dagger$ | $Re[\Phi_{4/5/6}]$ as $\overline{\psi}\tau^2 \otimes \sigma^{1/2/3}\psi$ |
| Honeycomb | $\Phi_+^\dagger$ | $e^{-i\frac{2\pi}{3}}\Phi_+$ | $e^{-i\frac{2\pi}{3}}\Phi_+$ | $\Phi_+$ | $e^{i\frac{\pi}{3}}\Phi_-^\dagger$ | $\Phi_+$ | $Im[\Phi_1]$ as $\overline{\psi}\tau^1\psi$ |
| Honeycomb | $\Phi_-^\dagger$ | $e^{\frac{2\pi}{3}}\Phi_-^\dagger$ | $e^{\frac{2\pi}{3}}\Phi_-^\dagger$ | $\Phi_-$ | $e^{\frac{\pi}{3}}\Phi_+^\dagger$ | $\Phi_-$ | $Im[\Phi_2]$ as $\overline{\psi}\tau^2\psi$ |
| Honeycomb | $\Phi_3^\dagger$ | $\Phi_3^\dagger$ | $\Phi_3^\dagger$ | $\Phi_3$ | $\Phi_3^\dagger$ | $\Phi_3$ | $Re[\Phi_3]$ trivial |
| Honeycomb | $\Phi_{4/5/6}^\dagger$ | $\Phi_{4/5/6}^\dagger$ | $\Phi_{4/5/6}^\dagger$ | $-\Phi_{4/5/6}$ | $-\Phi_{4/5/6}^\dagger$ | $-\Phi_{4/5/6}$ | $Im[\Phi_{4/5/6}]$ as $\overline{\psi}\tau^3 \otimes \sigma^{1/2/3}\psi$ |

Symmetry operations $T_{1/2}$, $R_x$ denote translation along two lattice vectors (for honeycomb $T_{1/2}$ direction has $2\pi/3$ angle between them) and reflection along horizontal bonds, respectively. Rotation implies site centered four-fold rotation for the square lattice and hexagon centered six-fold rotation for the honeycomb. There is always a trivial monopole (highlighted in bold) for DSLs on both these bipartite lattices. On proliferating the trivial monopole the emergent symmetry is reduced from $U(1)_{top} \times SO(6) \to SO(5)$, and the 15 $SO(6)$ adjoint fermion bilinears spilt according to $5 + 10$. The five fermion bilinears, which form an $SO(5)$ vector, are now symmetry equivalent to five monopoles, as listed in the last column, which is relevant to the chiral symmetry breaking pattern described in the discussion of Eq. (10)

translation, rotation, and reflection) and time-reversal symmetries. The enlarged symmetry (such as $SO(6) \times U(1)_{top}/\mathbb{Z}_2$) will emerge at low energy if terms breaking this symmetry down to the microscopic symmetries are irrelevant. We will discuss this in detail in the following subsection.

A key question addressed in this paper is: given a $U(1)$ Dirac spin liquid realized on a lattice, how do monopoles transform under the microscopic symmetries? In other words, how are the microscopic symmetries embedded into the enlarged symmetry group? Clearly spin-rotation, acting on all gauge-invariant local operators as an $SO(3)$ group, can only be embedded as an $SO(3)$ subgroup of the $SO(6)$ flavor group, meaning that three of the six monopoles form a spin-1 vector, and the other three are spin singlets. Denote the three singlet monopoles as $\mathcal{V}_{1,2,3} = \Phi_{1,2,3}$ and the three spin-1 monopoles as $\mathcal{S}_{1,2,3} = \Phi_{4,5,6}$. In terms of filling zero modes these operators can be written as

$$
\begin{aligned}
\mathcal{V}_{1,2,3}^\dagger &= [\varepsilon_\tau \tau^{1,2,3}]^{\alpha\beta} \varepsilon_\sigma^{ss'} f_{\alpha,s}^\dagger f_{\beta,s'}^\dagger \mathcal{M}_{\text{bare}}^\dagger \\
\mathcal{S}_{1,2,3}^\dagger &= i\varepsilon_\tau^{\alpha\beta} [\varepsilon_\sigma \sigma^{1,2,3}]^{ss'} f_{\alpha,s}^\dagger f_{\beta,s'}^\dagger \mathcal{M}_{\text{bare}}^\dagger
\end{aligned}
\tag{6}
$$

where $\varepsilon$ refers to the $2 \times 2$ antisymmetric matrix, $\sigma$, $\tau$ corresponds to the spin and valley index and we have split the subscript in $f_i$ to valley indices $\alpha$, $\beta$, and spin $s$, $s'$. An important observation here is that since monopoles are local operators, they transform as linear representations of the symmetry group, in contrast to gauge charged fermions that transform under a projective symmetry group. Thus, for example, the monopoles transform as integer spin representations, unlike the spinons which carry spin one half.

Other discrete symmetries can be realized, in general, as combinations of certain $SO(6)$ rotations followed by a nontrivial $U(1)_{top}$ rotation, and possibly some combinations of $\mathcal{C}_0, \mathcal{T}_0, \mathcal{R}_0$. (Remember that Lorentz group acts trivially on the $2\pi$-monopoles.) Many of these group elements can be fixed from the symmetry transformations of the Dirac fermions $\psi_i$. For example, if the symmetry operation acts on $\psi$ as $\psi \to U\psi$ with a nontrivial $U \in SU(4)$, then we know that the monopoles should also be multiplied by an $SO(6)$ matrix $O$ that corresponds to $U$. This $SO(6)$ matrix $O$ can be uniquely identified up to an overall sign, which can also be viewed as a $\pi$-rotation in $U(1)_{top}$. The same logic applies to other operations including $\mathcal{C}, \mathcal{T}, \mathcal{R}$. The only exception is the flux symmetry $U(1)_{top}$: there is no information regarding $U(1)_{top}$ in the symmetry transformation properties of the low-energy Dirac fermions. Fixing the possible $U(1)_{top}$ rotations in the implementations of the microscopic

lattice symmetries, and exploring their consequences, is our main task.

Since this is a key point, let us expand on it. Recall that via the state-operator correspondence, the monopole operators can be constructed by filling the Dirac sea together with half of the zero modes. Under a lattice symmetry, the transformation of zero modes corresponds to the $SO(6)$ matrix, while the phase factor of $U(1)_{top}$ can be attributed to a Berry phase contribution, arising from the filled Dirac sea of the gauge charged partons. Physically, the $U(1)_{top}$ rotation arises from moving the monopole operator around a closed path that encloses gauge charge. This also inspires the numerical extraction of monopole quantum numbers as we will describe in the "Methods" section[36,38]. We remark that numerical calculations cannot determine monopole transformations under time reversal or reflections, while analytical methods reported in a parallel work[49] completely fix all transformations and reproduce the numerical results.

**Monopole symmetry actions summary and stability of the DSL.** The transformation properties of monopoles on various lattices are summarized in the three tables below. The fact that monopoles are local operators implies that they transform as linear (rather than projective) representations of the symmetry groups. Note, the bipartite lattices have a trivial monopole, i.e. one that transforms as the identity representation under all symmetries. We elaborate on the consequences of this observation below.

In Table 1, the monopole transformation properties for the honeycomb DSL and square lattice staggered flux DSL are presented. Results for the square lattice align with the results of $M$ transformations of ref. [38] after making the identification $\Phi_{1/3} = M_{3/2}, \Phi_2 = iM_1, \Phi_4 \mp i\Phi_5 = M_{4/6}, \Phi_6 = M_5$. For the special case where the flux on the square lattice is $\phi = \pi$ an additional unitary symmetry, charge conjugation, is present which sends $\Phi_{1/3/4/5/6}^\dagger \to \Phi_{1/3/4/5/6}$, and $\Phi_2^\dagger \to -\Phi_2$. Combining this with $T_{1/2}, C_4, \mathcal{T}$ gives the transformations for translations and rotations, time-reversal in this state.

The stability of the $U(1)$ Dirac spin liquid can be discussed in three stages. First, there is Eq. (3), $QED_3$ with $N_f = 4$ flavors, which neglects monopoles and has a global $SU(4)$ flavor symmetry. The stability of this theory has been discussed both from the numerical (lattice gauge theory) perspective[45,50] as well as from the epsilon expansion,[51] all of which conclude that it flows to a stable fixed point. We therefore begin our discussion by analyzing the remaining two effects—that of four fermion

**Table 2 Triangular lattice: fermion bilinears and monopole symmetries**

| | $T_1$ | $T_2$ | $R$ | $C_6$ | $\mathcal{T}$ |
|---|---|---|---|---|---|
| $M_{00}$ | + | + | − | + | − |
| $M_{i0}$ | + | + | + | − | + |
| $M_{01}$ | − | − | $-M_{03}$ | $-M_{02}$ | + |
| $M_{02}$ | + | − | $M_{02}$ | $M_{03}$ | + |
| $M_{03}$ | − | + | $-M_{01}$ | $M_{01}$ | + |
| $M_{i1}$ | − | − | $M_{i3}$ | $M_{i2}$ | − |
| $M_{i2}$ | + | − | $-M_{i2}$ | $-M_{i3}$ | − |
| $M_{i3}$ | − | + | $M_{i1}$ | $-M_{i1}$ | − |
| $\Phi_1^\dagger$ | $e^{-i\frac{\pi}{3}}\Phi_1^\dagger$ | $e^{i\frac{\pi}{3}}\Phi_1^\dagger$ | $-\Phi_3^\dagger$ | $\Phi_2$ | $\Phi_1$ |
| $\Phi_2^\dagger$ | $e^{i\frac{2\pi}{3}}\Phi_2^\dagger$ | $e^{i\frac{\pi}{3}}\Phi_2^\dagger$ | $\Phi_2^\dagger$ | $-\Phi_3$ | $\Phi_2$ |
| $\Phi_3^\dagger$ | $e^{-i\frac{\pi}{3}}\Phi_3^\dagger$ | $e^{-i\frac{2\pi}{3}}\Phi_3^\dagger$ | $-\Phi_1^\dagger$ | $-\Phi_1$ | $\Phi_3$ |
| $\Phi_{4/5/6}^\dagger$ | $e^{i\frac{2\pi}{3}}\Phi_{4/5/6}^\dagger$ | $e^{-i\frac{2\pi}{3}}\Phi_{4/5/6}^\dagger$ | $\Phi_{4/5/6}^\dagger$ | $-\Phi_{4/5/6}$ | $-\Phi_{4/5/6}$ |

The $M_{ij} = \bar{\psi}\sigma^i\tau^j\psi$ denotes the 16 fermion mass terms. Their transformation under lattice and time reversal symmetry are shown followed by the corresponding table for the six magnetic monopoles $\Phi_i$. Symmetries $T_{1/2}$, $R$, $C_6$ denote translation and reflection marked in Fig. 1, and six-fold rotation around a site, respectively

**Table 3 Fermion bilinear and monopole symmetries on the kagome lattice**

| | $T_1$ | $T_2$ | $R_y$ | $C_6$ | $\mathcal{T}$ |
|---|---|---|---|---|---|
| $M_{00}$ | + | + | − | + | − |
| $M_{01}$ | − | − | $-M_{03}$ | $M_{02}$ | + |
| $M_{02}$ | + | − | $M_{02}$ | $-M_{03}$ | + |
| $M_{03}$ | − | + | $-M_{01}$ | $-M_{01}$ | + |
| $M_{i0}$ | + | + | − | + | + |
| $M_{i1}$ | − | − | $-M_{i3}$ | $M_{i2}$ | − |
| $M_{i2}$ | + | − | $M_{i2}$ | $-M_{i3}$ | − |
| $M_{i3}$ | − | + | $-M_{i1}$ | $-M_{i1}$ | − |
| $\Phi_1^\dagger$ | $-\Phi_1^\dagger$ | $-\Phi_1^\dagger$ | $-\Phi_3$ | $e^{i\frac{2\pi}{3}}\Phi_2^\dagger$ | $\Phi_1$ |
| $\Phi_2^\dagger$ | $\Phi_2^\dagger$ | $-\Phi_2^\dagger$ | $\Phi_2$ | $-e^{-i\frac{2\pi}{3}}\Phi_3^\dagger$ | $\Phi_2$ |
| $\Phi_3^\dagger$ | $-\Phi_3^\dagger$ | $\Phi_3^\dagger$ | $-\Phi_1$ | $-e^{-i\frac{2\pi}{3}}\Phi_1^\dagger$ | $\Phi_3$ |
| $\Phi_{4/5/6}^\dagger$ | $\Phi_{4/5/6}^\dagger$ | $\Phi_{4/5/6}^\dagger$ | $-\Phi_{4/5/6}$ | $e^{i\frac{2\pi}{3}}\Phi_{4/5/6}^\dagger$ | $-\Phi_{4/5/6}$ |

Symmetry transformation of fermion bilinears and monopoles on the kagome lattice, where $M_{ij} \equiv \bar{\psi}\sigma^i\tau^j\psi$. Translations are marked in Fig. 1. $R_y$, $C_6$ denotes reflection with respect to $y$-axis and six-fold rotation around center of hexagon. The six-fold rotation symmetry acting on monopoles cannot be incorporated into the vector representation of SO(6) owing to the nontrivial Berry phase, which is in line with the magnetic pattern expected on the kagome lattice

interaction terms that break SU(4) symmetry, and that of magnetic monopoles.

Let us begin with the scaling dimension of the monopole operator. Within a large $N_f$ approximation, the scaling dimension is: $\Delta_1 = 0.265 N_f - 0.0383 + O(1/N_f)$, so setting $N_f = 4$ yields $\Delta_1 = 1.02 < 3$, which implies that this operator is strongly relevant. While the true scaling dimension at $N_f = 4$ could be different, this is unlikely to exceed 3. We will therefore assume that the single monopole operator is a relevant perturbation. For the bipartite lattices, the presence of a trivial monopole implies a single monopole insertion operator is allowed on symmetry grounds in the Lagrangian. Then, we do not expect the $U(1)$ Dirac spin liquid to be a stable phase. What does it flow to? The most likely scenario is that chiral symmetry is broken, i.e. a mass term is developed by spontaneous symmetry breaking. This still leaves a gapless photon, which is removed by monopole proliferation[52]. We will argue below that this does not lead to additional symmetry breaking, and conclude that the colinear Neel order or common VBS orders on bipartite lattices are likely to be realized in this theory at the lowest energies.

On the other hand for the non-bipartite lattice DSLs considered here, i.e. the triangular and kagome DSL, no such trivial monopole is present. This has a number of consequences. First, the QED$_3$ theory discussed here could potentially represent a stable phase, with an enlarged $SU(4) \times U(1)/Z_4$ global symmetry which appears in the low-energy limit. We discuss this and other possibilities below. First, let us discuss the issue of monopole operator scaling dimensions. For the triangular lattice, under translations (see Table 2), note that $\Phi_{1,2}$ have $k_1 = \pi/3$ and $\Phi_3$ has $k_1 = -2\pi/3$, and the lowest order invariant monopole terms are:

$$\Delta\mathcal{L}_{\text{Triangular}} = \Phi_1\Phi_2\Phi_3 + \text{h.c.} \tag{7}$$

Note, the mismatch in momentum with fermion bilinears, which only pick up phase factors that are multiples of $\pi$, implies that there is no invariant term with a smaller monopole charge. Although zero modes for $6\pi$ (three-fold) monopole as in Eq. (7) carries Lorentz spin-1, the leading-order three-fold monopoles contain lorentz singlet ones and transform formally as the above term, detailed construction contained in Supplementary Note 4. Within a large $N_f$ calculation[53], the scaling dimension of this triple monopole is $\Delta_3 = 1.186 N_f - 0.422 + O(1/N_f) \sim 4.32$, which makes it very likely to be an irrelevant perturbation at the SU(4) symmetric fixed point. The remaining operator to inspect is the

four fermion that breaks SU(4) symmetry, that can be written as $\mathcal{L}_4 = \sum_{a=1}^{3}(\bar{\psi}\sigma^a\psi)^2 - (\bar{\psi}\tau^a\psi)^2$. While this operator is irrelevant at tree level, interactions could change its scaling dimension. A recent epsilon expansion study[51] reports the scaling dimension of this operator to be $\Delta_{4f} = 3.17$, which means it would remain irrelevant, although significant uncertainty is associated with this scaling dimension, and other approximations, such as large $N_f$, imply that it is relevant[35]. This would decide whether the Dirac spin liquid a stable phase, with no relevant operators, or a critical point with a single relevant operator, which would require tuning of the four fermion term $\mathcal{L}_4$ to access[35]. In either case it is expected to be relevant to understanding the phase structure on the triangular lattice.

In contrast, on the kagome lattice an inspection of the monopole and mass term transformation laws imply (see Table 3) the following two invariant terms:

$$\Delta\mathcal{L}_{\text{kagome}}^1 = M_{01}\left(\Phi_1 e^{i\frac{2\pi}{3}}\right) + M_{02}\left(\Phi_2\right) + M_{03}\left(\Phi_3 e^{-i\frac{2\pi}{3}}\right) + \text{h.c.}$$
$$\Delta\mathcal{L}_{\text{kagome}}^2 = e^{i\frac{2\pi}{3}}(\Phi_1^\dagger)^2 + (\Phi_2^\dagger)^2 + e^{-i\frac{2\pi}{3}}(\Phi_3^\dagger)^2 + \text{h.c.}$$
$$\tag{8}$$

where $M_{0i} \equiv \bar{\psi}\tau^i\psi$. Note, the first term involves a combination of a single monopole insertion operator and a fermion bilinear, which may be regarded as the excited state of a monopole with larger scaling dimension, and the second term refers to doubled monopole insertion and preserves symmetry if considering the associated lorentz singlet operator, details in Supplementary Note 4. The scaling dimensions for these operators, estimated from large $N_f$ is $\Delta_{1*} = \Delta_1 + 2\sqrt{2} \sim 3.84$ and $\Delta_{2*} = 0.673 N_f - 0.194 \sim 2.50$. While the second one is nominally relevant, their closeness to 3 implies that we should leave open the possibility of a stable phase or critical point on the kagome lattice described by a $U(1)$ Dirac spin liquid. Regardless of stability, this difference in the nature of the monopoles from the bipartite case will have an important impact on proximate orders that we document below. In particular, relatively complex magnetic orders such as the 120° state and the 12 site VBS pattern on the triangular lattice are captured.

**Symmetry breaking, monopole proliferation, and ordered states.** Now, we will be concerned with identifying ordered states that can be reached from the Dirac spin liquid, either as a result of

an intrinsic instability, or because interactions are tuned to trigger a phase transition. The scenario that we will assume is that of a two step process with spontaneous mass generation occurring first, i.e. a fermion bilinear spontaneously acquires an expectation value by symmetry breaking, followed by the monopole proliferation and confinement[52]. This particular ordering of condensation is not universal but rather provides a controlled limit for our purpose. The 16 fermion bilinears are classified as $1 \oplus 15$, a singlet and adjoint representation of SU(4)~SO(6). Depending on the symmetries of the interaction, a mass term $\bar{\psi} \boldsymbol{M} \psi$, with $\boldsymbol{M}$ being either the identity or a vector such as $\boldsymbol{M} = (M_{01}, M_{02}, M_{03})$, can be generated. This is captured by the following Gross–Neveu type model:

$$\mathcal{L} = \sum_{i=1}^{4} \bar{\psi}_i i \not{D}_a \psi_i + g\phi \cdot \bar{\psi} \boldsymbol{M} \psi + (\partial_\mu \phi)^2 - u\phi^2 - \lambda\phi^4, \quad (9)$$

$\phi$ represents bosonic fields, which can either be a scalar field or a vector field depending on the type of generated mass $\bar{\psi} \boldsymbol{M} \psi$.

The singlet mass is a quantum Hall mass term $\bar{\psi}\psi$, which breaks time reversal and parity symmetry. If spontaneously generated, it will lead to a chiral spin liquid, a gapped phase with topological order but gapless edge states and semion excitations. In this scenario, the Chern Simons term suppresses monopole proliferation.

The second scenario is when chiral symmetry is broken by the spontaneous generation of one of the 15 chiral mass terms, which are conveniently labeled in terms of $M_{i0} = \bar{\psi}\sigma^i \otimes 1\psi$; $M_{0j} = \bar{\psi}1 \otimes \tau^j \psi$ and $M_{ij} = \bar{\psi}\sigma^i \otimes \tau^j \psi$. Different from the quantum Hall (singlet) mass $\bar{\psi}\psi$, the chiral mass does not lead to a Chern–Simons term. Therefore, we are left with a pure $U(1)$ gauge theory, which may have a further instability to monopole proliferation and confinement. A key input is to identify which monopole is selected following the chiral symmetry breaking by one of the 15 mass terms. Operationally, this selection arises since the mass term splits the zero mode degeneracy in the monopole. For example, consider a 'quantum spin Hall' mass term $M_{30} = \bar{\psi}\sigma^3 \otimes 1\psi$, that associates $S_z$ spin density with magnetic flux. A magnetic monopole then has both the spin-down zero modes filled, corresponds to $\mathcal{S}_1^\dagger + i\mathcal{S}_2^\dagger$, which, when inserted into Eq. (6) yields $\varepsilon_\tau^{\alpha\beta}[\sigma^z - 1]^{ss'} f_{\alpha,s}^\dagger f_{\beta,s'}^\dagger \mathcal{M}_{\text{bare}}^\dagger$, consistent with their filling down spin modes. A general relation between the mass terms and monopoles can be obtained by doing a SO(6) rotation on the familiar case of quantum spin Hall mass. (The mass terms are in the adjoint representation of SO(6) and the monopoles are the SO(6) vectors.) We find that in general, a mass term $\pm\bar{\psi}T^{ab}\psi$ will lead to proliferation of the monopole $(\hat{n}_a \pm i\hat{n}_b) \cdot \Phi$, where $T^{ab}$ is the SO(6) generator that rotates in the plane spanned by the two orthogonal unit vectors $\{\hat{n}_a, \hat{n}_b\}$. Specifically,

1. The mass term $\pm M_{c0}$ will proliferate the $\mathcal{S}_a \pm i\mathcal{S}_b$ monopole, where $(a, b, c)$ is an even permutation of $(1, 2, 3)$; this leads to non-collinear magnetic order that fully breaks $SO(3)_{\text{spin}}$, see the subsections treating the triangular and kagome lattices below, and Supplementary Note 5.
2. The mass term $\pm M_{0c}$ will proliferate the $\mathcal{V}_a \pm i\mathcal{V}_b$ monopole, where $(a, b, c)$ is an even permutation equivalent of $(1, 2, 3)$; this leads to valence bond solid (VBS) type order that breaks lattice symmetries, as we discuss in the remainder of this section.
3. The mass term $\pm M_{ab}$ will proliferate the $\mathcal{S}_a \mp i\mathcal{V}_b$ monopole, resulting in mixed order with collinear spin order along $\sigma^a$ direction and VBS order, see Supplementary Note 5, with the exception of pure Neel order for mass $M_{a2}$, $M_{a3}$ on square and honeycomb lattices, respectively, see the following subsection.

**Bipartite lattices**. We now discuss in more detail the consequence of a trivial monopole on the bipartite lattices, where chiral symmetry breaking alone may determine the ordered states, in contrast to the triangular and kagome cases where the monopole proliferation (confinement) leads to additional symmetry breaking.

We first highlight the existence of a trivial monopole in the DSL on bipartite lattices shown in Table 1. A trivial monopole, by our definition, stays invariant (or goes to its conjugate) under translations, rotations, reflections, and time-reversal symmetry. (Note that time-reversal reverses the monopole charge, so by invariant we mean $\Phi \to \Phi^\dagger$ instead of $\Phi \to -\Phi^\dagger$).

The presence of the trivial monopole in the Lagrangian will presumably drive the theory to strong coupling. The emergent symmetry is broken from $SO(6) \times U(1)/\mathbb{Z}_2$ to $SO(5)$ by this trivial monopole. The remaining $SO(5)$ still possess a symmetry anomaly[44], so the theory cannot flow to a trivially gapped symmetric state. A natural assumption is that in the absence of further fine tuning, this will lead to chiral symmetry breaking. But the 15 adjoint masses no longer stand on equal footing: There is a term allowed in Lagrangian coupling fermion bilinears and monopoles that is invariant under $SO(6) \times U(1)_{\text{top}}/\mathbb{Z}_2$ and other discrete symmetries.

$$\begin{aligned}\mathcal{L}_{\text{mass-mon}} = \ &- \sum_{i=1,2,3} \epsilon^{ijk}[M_{i0}(i\mathcal{S}_j^\dagger \mathcal{S}_k) + M_{0i}(i\mathcal{V}_j^\dagger \mathcal{V}_k)] \\ &+ \sum_{i,j=1,2,3} M_{ij}(i\mathcal{S}_i^\dagger \mathcal{V}_j + \text{h.c.})\end{aligned} \quad (10)$$

Once the trivial monopole $\mathcal{V}_j$ condenses, the equation above picks out the terms involving $\mathcal{V}_j^{(\dagger)}$ which then becomes more relevant than other mass terms—hence the masses $M_{0i}(i \neq j)$, $M_{aj}(a = 1, 2, 3)$ are more likely to be generated in chiral symmetry breaking. These five mass terms form a vector under the $SO(5)$ flavor symmetry that remains unbroken by the $\mathcal{V}_j$ condensation, while the remaining (less relevant) 10 mass terms transform as a rank-2 symmetric traceless tensor.

For example, on the square lattice, there is a symmetry trivial monopole $i\mathcal{V}_2 - i\mathcal{V}_2^\dagger = 2\,\text{Im}\,\mathcal{V}_2$. Its proliferation will lead to the spontaneous generation of a mass term $M_i \in \{\bar{\psi}\tau^2 \otimes \sigma^i\psi, \bar{\psi}\tau^{1/3}\psi\}$ per discussion above, yielding spontaneous symmetry breaking. Indeed, the chiral symmetry breaking states are the familiar Neel or columnar VBS states. First, we note that the mass terms $\bar{\psi}\sigma^i \otimes \tau^2\psi$ have the same symmetries as the familiar Neel order along the $\sigma^i$ direction, while mass terms $\bar{\psi}\tau^{1,3}\psi$ have the same symmetry transformation properties as the columnar VBS order parameter (Supplementary Table 1 for mass transformation). According to our previous discussion, the generation of a mass term will further lead to monopole proliferation. This however does not further break symmetry. From Eq. (10) it is clear that in the presence of the trivial monopole $\mathcal{V}_j$, the coupling between $SO(5)$-vector mass terms and the five nontrivial monopoles becomes effectively linear (since $\langle \mathcal{V}_j \rangle \neq 0$), which makes the two sets of operators essentially identical from symmetry point of view. Therefore monopoles will not further break any symmetry after an $SO(5)$-vector mass condensate is established. For example, on the square lattice the mass term $\bar{\psi}\tau^2\sigma^3\psi$ will lead to the monopole condensation $\langle \text{Im}(\mathcal{V}_2 + i\mathcal{S}_3) \rangle \neq 0$, which does not break further symmetries. Similarly on honeycomb lattice, the five masses $\{\bar{\psi}\tau^3\sigma^i\psi, \bar{\psi}\tau^{1/2}\psi\}$ leads to proliferation of monopoles $\mathcal{V}_3 + i\mathcal{S}_i$, $\mathcal{V}_{2/1} + i\mathcal{V}_3$, respectively, which result in Neel/Kekule VBS. These orders also align with the corresponding masses (Supplementary Table 1 for mass transformation).

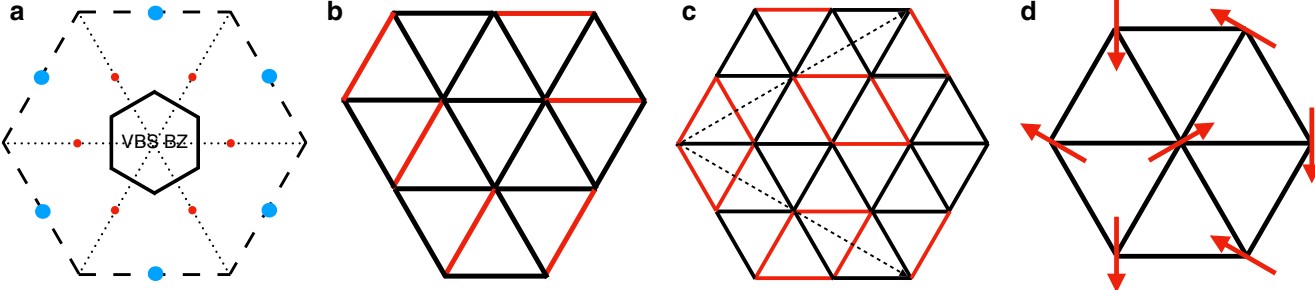

**Fig. 2** Triangular lattice VBS and magnetic orders. **a** The reduced briouilin zone (BZ) stipulated by the momenta of $\mathcal{V}_{1,2,3}$, $\bar{\psi}\tau^{1,2,3}\psi$ which has an area 1/12 of the original one. The red/blue dots mark momenta of $\mathcal{V}_i$'s and $\bar{\psi}\tau^{1,2,3}\psi$, respectively. **b** One particular VBS pattern inside a 12-site unit cell used in previous numerics for quantum dimer models which can result from any generic $\bar{\psi}\tau^i\psi$ and subsequent monopole proliferation, so long as they break all but the enlarged VBS Bravais lattice translations. Red/black bonds denote positive/vanishing valence bond weight. **c** A more symmetric VBS, invariant under $C_3$ around the blue dot. **d** The 120° magnetic order induced by mass $\bar{\psi}\sigma^3\psi(M_{30})$

Potentially, on tuning the balance between Neel and VBS orders, a deconfined critical point may be accessed[4,14,44] (the transition may also be first order, depending on the exact long distance fate of the theory). The possible emergent SO(5) symmetry[54] at the Neel-VBS transition is nothing but the unbroken flavor symmetry of QED$_3$. It is important to note that the SO(5) emergent symmetry of the deconfined critical point is only a subgroup of the $\sim U(4)$ symmetry of the DSL.

Previously, there has been debate regarding the ground state associate with the staggered flux spin liquid state on the square lattice. Based on the monopole quantum numbers reported here (and in ref. [38]) and the discussion above, we conclude that it describes an ordered state. Although the precise nature of the order is determined by microscopic interactions, it is certainly compatible with the commonly observed colinear Neel order.

If rotation symmetry is broken (e.g., from square to rectangle lattices), monopole momenta are not enforced to be quantized (see Supplementary Note 1) by algebraic relations and generally we expect all elementary monopoles are forbidden by symmetry. The stability of DSL on square and honeycomb lattices is hence enhanced.

**Triangular lattice**. On the triangular lattice, we find the symmetry transformation properties detailed in Table 2. We remark that the momenta of monopoles receive a contribution from the nontrivial Berry phase in $U(1)_\mathrm{top}$ for translations.

As we have noted, magnetic monopoles correspond to local operators, and their symmetry transformation allows us to interpret them as order parameters, which, if condensed, break symmetry in particular ways.

For magnetic orders, for example, the spin triplet monopoles $\mathcal{S}_{1,2,3}$ are the order parameter of the 120° non-collinear magnetic order. If the monopoles condense, we will have a spin ordering pattern

$$\langle \mathbf{S}_r \rangle = S(\mathbf{n}_1 \cos(\mathbf{Q} \cdot \mathbf{r}) + \mathbf{n}_2 \sin(\mathbf{Q} \cdot \mathbf{r})) \qquad (11)$$

with $\mathbf{n}_1 = (\mathrm{Re}[\mathcal{S}_1], \mathrm{Re}[\mathcal{S}_2], \mathrm{Re}[\mathcal{S}_3])$ and $\mathbf{n}_2 = (\mathrm{Im}[\mathcal{S}_1], \mathrm{Im}[\mathcal{S}_2], \mathrm{Im}[\mathcal{S}_3])$. $\mathbf{Q} = (2\pi/3, -2\pi/3)$ is the momentum of monopoles. To establish that this order is the 120° non-collinear magnetic order, we need to further show that monopoles condense in a channel that satisfies $\mathbf{n}_1 \cdot \mathbf{n}_2 = 0$. Recall that there are two steps to generate an order. First, a fermion mass is spontaneously generated through chiral symmetry breaking. Next, the mass term will pick one monopole to condense. This two-step mechanism guarantees $\mathbf{n}_1 \cdot \mathbf{n}_2 = 0$. For instance, the mass $\bar{\psi}\sigma^3\psi$ will have the monopoles condensing in the channel

$\langle \mathcal{S}_1 + i\mathcal{S}_2 \rangle \neq 0$, $\langle \mathcal{S}_1 - i\mathcal{S}_2 \rangle = 0$, and $\langle \mathcal{S}_3 \rangle = 0$, which satisfies the constraint $\mathbf{n}_1 \cdot \mathbf{n}_2 = 0$. This eventually yields the 120° magnetic order with the magnetic moments lying in the $S_x S_y$ plane, with the specific chirality shown in Fig. 2d.

The general principle for determining the VBS order induced by a specific chiral symmetry breaking scenario involves the following single rule: match only the preserved symmetries. By this we exclude symmetries under which the mass/monopoles obtain a nontrivial phase (i.e., only when they stay strictly invariant do the symmetries count as being preserved). When a valley-Hall mass $M_{0i}$ leads to the condensation of spin singlet monopoles $\mathcal{V}_j + i\mathcal{V}_k$ giving rise to VBS order, typically, the order parameter is a polynomial of the condensed monopole and the valley Hall mass. This polynomial retains only the common preserved symmetries of $\mathcal{V}_i$ and the mass, but not any nontrivial transformation (or phases) (e.g., under original translations); hence so does the VBS order. For example, consider translations. It suffices for the VBS pattern to have a new (and usually enlarged) unit cell commensurate with both the fermion bilinear and the condensed monopole, i.e., nontrivial phases under translations on the original lattice do not further constrain the VBS order. This is different for the case of spin order associated with the $\mathcal{S}_i$'s. There the nontrivial Berry phase $e^{i2\pi/3}$ associated with lattice translations implies that lattice translations combined with some $2\pi/3$ spin rotations (along the axis defined by the quantum spin Hall mass) are kept unbroken. Another unbroken symmetry associated with the magnetic order is the combination of time-reversal and a spin $\pi$-rotation along the axis defined by the quantum spin Hall mass—this symmetry implies that the magnetic order is co-planar. In fact these unbroken symmetries essentially fix the form of the 120° order.

Following this logic, the new Bravais lattice for VBS on triangular lattice is set by the momenta of $\Phi_{1/2/3}$ and $\bar{\psi}\tau^i\psi$. So from Fig. 2a, the reduced Brillouin zone has an area which is $\frac{1}{12}$ of the original one resulting from the Berry phase attached to the monopoles, corresponding exactly to the $\sqrt{12} \times \sqrt{12}$ VBS order with a 12-site unit cell. A generic combination of $\bar{\psi}\tau^i\psi$ induces monopole condensation that breaks all spatial symmetries except for translations commensurate with the new unit cell. Following the previous discussion, there are no further symmetry constraints on the VBS order pattern inside the enlarged unit cell. In the following, we display some typical patterns that could descend from the DSL. Shown in Fig. 2b is a generic pattern used in ref. [55] as a ground state candidate for a quantum dimer model on triangular lattice, with maximal flippable plaquettes. The plaquette VBS shown in Fig. 2c has an additional $C_3$ symmetry

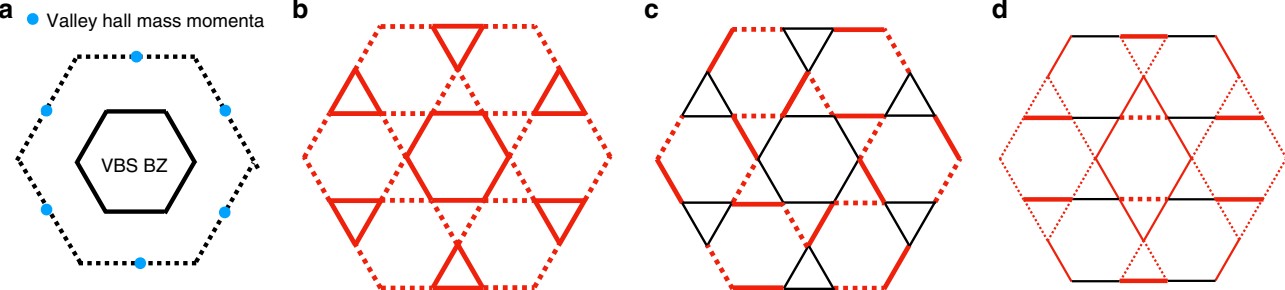

**Fig. 3** Kagome lattice VBS. **a** The momenta of $M_{0i}$ and $\mathcal{V}_i$'s on kagome lattice yielding the reduced Brillouin zone for VBS pattern with an area a quarter of the original one. Panel **b**–**d** are examples of high-symmetry VBS patterns on kagome lattice resulting from chiral symmetry breaking and monopole proliferation. The solid/dotted red bonds denotes positive/negative VBS weight with width indicating strength. Weight vanishes on black bonds. Patterns **b** and **c** result both from condensation of mass $M_{01} + M_{02} - M_{03}$, but with the corresponding monopole $\Phi^\dagger_{\text{kag}} = \mathcal{V}^\dagger_1 e^{-i\frac{2\pi}{3}} + \mathcal{V}^\dagger_2 - \mathcal{V}^\dagger_3 e^{i\frac{2\pi}{3}}$ condensing to $\langle\Phi^\dagger_{\text{kag}}\rangle = 1, i$, respectively. Pattern **d** results from mass $M_{02}$ with $\langle\mathcal{V}^\dagger_1 + i\mathcal{V}^\dagger_3\rangle = e^{-i\frac{\pi}{4}}$, which preserves $R_y$ but breaks $C_6$. Pattern **b** is symmetry equivalent to the Hastings VBS of ref. [24], and Pattern **d** is symmetry equivalent to the VBS found by DMRG proximate to the spin liquid phase in ref. [56] (panel **c** of Fig. 2)

and results from a particular mass $M_{01} + M_{02} - M_{03}$ with the corresponding $\mathcal{V}^\dagger_1 + \mathcal{V}^\dagger_2 e^{i\frac{\pi}{3}} + \mathcal{V}^\dagger_3 e^{i\frac{2\pi}{3}}$ proliferation, both of which stay invariant under $C_3$ around the marked triangle center (equivalently $T_1 C^2_6$) according to Table 2.

**Kagome lattice**. On the kagome lattice, the situation is very similar to the triangular lattice, where the Berry phase of Dirac sea for $C_6$ rotation is $\frac{2\pi}{3}$ by numerical calculation (Supplementary Note 3), which is consistent with the translationally invariant ('q = 0') 120° magnetic order under six-fold rotation. The results are summarized in Table 2.

Masses $M_{0i}$ result in VBS order. Generally, since the momenta of masses and monopoles are located at points like $(0, \pi)$ in the brillouin zone (leftmost panel of Fig. 3, the VBS pattern has an enlarged unit cell four times as big as the original one.

As before, we construct VBS order parameters from chiral symmetry breaking by looking at only the symmetries strictly preserved by both the mass and the condensed monopole. Shown in Fig. 3 are examples which reproduce previously found VBS patterns on the kagome lattice[28,56,57]. Specifically, Fig. 3(c) results from mass $M_{02}$ with $\langle\mathcal{V}^\dagger_1 + i\mathcal{V}^\dagger_3\rangle = e^{-i\frac{\pi}{4}}$, which preserves $R_y$ but breaks $C_6$ according to Table 3; the bond patterns have the same symmetry group (translations and $R_y$) as that found in a recent DMRG study proximate the spin liquid phase (panel c of Fig. 2 in ref. [56]). Further, Fig. 3(a) (b) result from a $C_6$ invariant mass $M_{01} + M_{02} - M_{03}$ and the associated monopole $\langle\Phi^\dagger_{\text{kag}}\rangle = \langle\mathcal{V}^\dagger_1 e^{-i\frac{2\pi}{3}} + \mathcal{V}^\dagger_2 - \mathcal{V}^\dagger_3 e^{i\frac{2\pi}{3}}\rangle = 1, i$, respectively. They are $C_6$ invariant, while (1) preserves $R_y$ and (2) breaks $R_y$, owing to $\Phi^\dagger_{\text{kag}} \to \Phi_{\text{kag}}$ under $R_y$, leading to the preservation of $R_y$ if $\langle\Phi^\dagger_{\text{kag}}\rangle = 1$ in 1. Figure 3(a) reproduces Fig. 4 in ref. [24] or Fig. 5 in ref. [28], where the 12 bonds around a unit cell are enhanced (or weakened an in our convention). Moreover symmetries of Figs. 17 and 18 in ref. [28] both align with the real part of $\mathcal{V}_i$'s. We remark that the specific expectation values of monopoles discussed above sit at the extremum of Eq. (8) and hence these more symmetric patterns optimize the Landau-potential given by the two-fold monopole on kagome lattice.

## Discussion

Our calculation of symmetry quantum numbers of monopole excitations in the QED$_3$ Dirac spin liquid theory on different 2D lattices indicates that the DSL on triangular and kagome lattices may be a stable phase. This would represent a remarkable state of matter, with enhanced symmetries including a $\sim U(4)$ symmetry

combining spin rotation and discrete spatial symmetries, as well as invariance under conformal (including scaling) transformations. We emphasize that the low-energy excitations of the DSL includes both fermionic spinons and the magnetic monopole continuum. These two types of excitations have different characteristic signatures in the spectral function, originating from the different scaling dimensions of the operators. They are also typically located at different high symmetry points of the Brillouin zone. The low-energy spin-triplet excitation arising from pairs of spinons are located near the $M$ points of the brillouin zone for both the triangular and kagome lattice. In contrast, the spin-triplet monopole excitations appear at the $K$ points for the triangular lattice, and at the $\Gamma$ point for the kagome lattice. One should note that there is always a spin-triplet mode at zero momentum on any lattice due to the conservation of total spin. However this mode has scaling dimension $\Delta = 2$, and is expected to be less singular than the monopole modes described here. These readily measurable characteristics should help in the empirical search for these phases.

Comparing with numerical studies, the kagome Heisenberg antiferromagnet (KHA) was recently argued to be consistent with a Dirac spin liquid[28,58,59]. Indeed, on increasing the next neighbor coupling $J_2 > 0$, a 120° (the so called q = 0) ordered phase is observed on exiting the spin liquid. Furthermore, a diamond VBS pattern (Fig. 3(d)) was found to be proximate the spin liquid of KHA[56], consistent with our identification of proximate orders. A third piece of evidence is that a chiral spin liquid (CSL) is observed by increasing the $J_2$ and $J_3$ interactions[60–62], or by adding a small spin chirality term $J_\chi S_i \cdot (S_j \times S_k)$[63]. This CSL can be understood as the DSL with a singlet mass $m\bar\psi\psi$.

It should however be noted that gapped spin liquids have also been proposed as the ground state in this parameter regime[56,64–66]. In Herbertsmithite, a material realization of the KHA, a spin liquid ground state is reported. Although low-energy excitations are observed, an analysis that includes disorder points to a spin liquid with a small spin gap of $\sim J/20$[67,68]. Nevertheless, a comparison of the low-energy spectral weight in kagome materials with the predictions of the DSL would be a useful exercise. In the left column of Fig. 4 we list the momentum and the reflection eigenvalue of 15 fermion adjoint masses (with identical scaling dimension) and 1 singlet mass, as well as the momenta of monopoles and the angular momentum of $\mathcal{V}_i$'s.

For the $S = 1/2$ triangular lattice antiferromagnet, numerical diagonalization and density matrix renormalization group (DMRG) studies have revealed a spin-disordered state on adding a small second neighbor coupling $0.07 < J_2/J_1 < 0.15$[69]. Variational Monte Carlo calculations have concluded that the Dirac spin

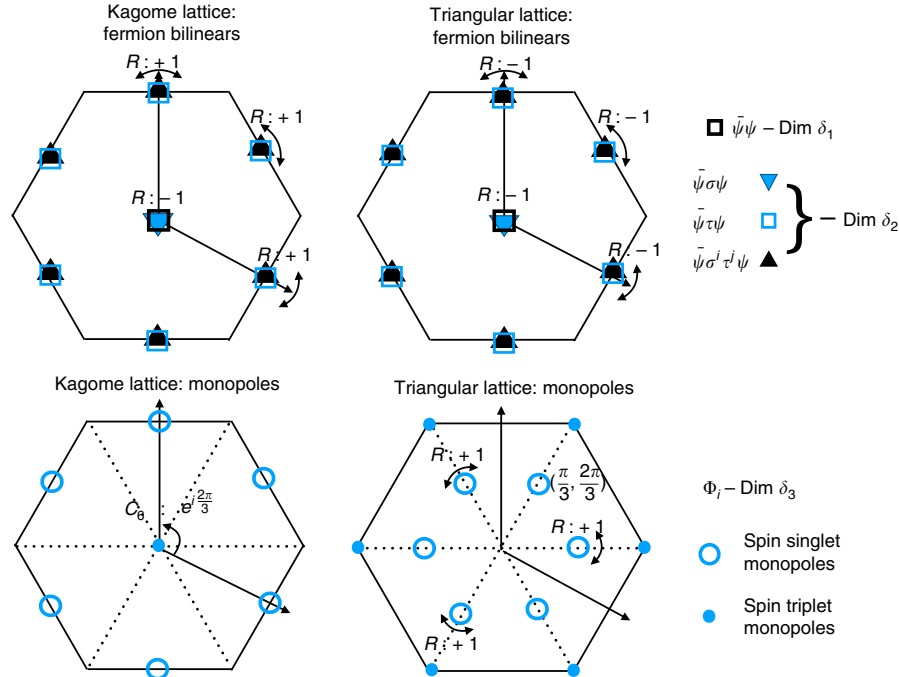

**Fig. 4** Symmetry quantum numbers of dominant operators of the DSL on kagome and triangular lattices Fermion bilinears and monopoles lead to measurable characteristic signatures in numerics and scattering experiments. These include the $1 + 15$ fermion bilinears and the six monopoles. In addition to translation quantum numbers (crystal momenta), rotation and reflection eigenvalues (where applicable) are also shown (reflection eigenvalues for bilinears at $\Gamma$ point refer to all three reflection operators marked in the figure; note for triangular bilinears, reflection used is different from that in Table 2 or Fig. 1). The spin triplet monopoles on the kagome lattice have angular momentum $l = 2$ under $C_6$. For triangular lattice monopoles, reflection eigenvalues apply to both spin singlet and triplet monopoles. The fermion bilinears and monopoles have different scaling dimensions as shown in the figure with the current best estimates indicating $\delta_1 > \delta_2 > \delta_3$

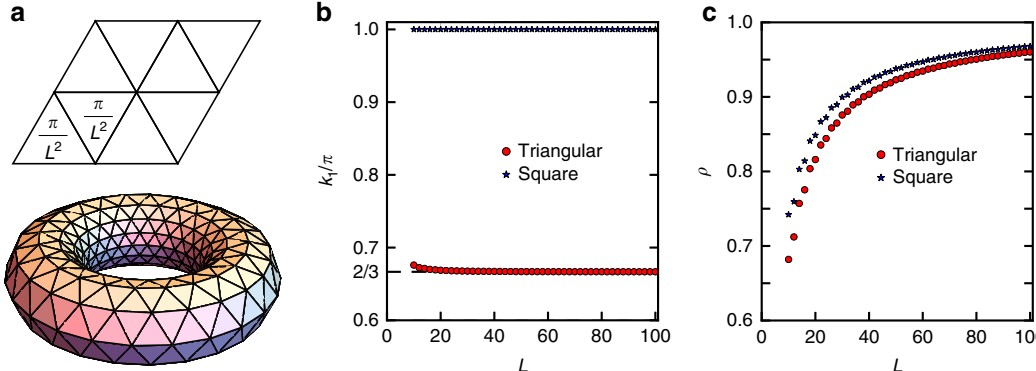

**Fig. 5** Numerical results of momentum of the monopoles. **a** We wrap the system on a $L \times L$ torus, and uniformly spread $2\pi/L^2$ flux in each unit cell. **b, c** To extract the lattice momentum, we calculate $\langle \psi | G_{T_1} \cdot T_1 | \psi \rangle = \rho e^{ik_1}$, and find $k_1 = 2\pi/3$ on the triangular lattice and $k_1 = \pi$ on the square lattice. $\rho$ is close to 1 for a large system size, which indicates that the translation symmetry is approximately preserved

liquid[33] is a very competitive ground state. A further piece of evidence is obtained on adding an explicit but small spin chirality interaction $J_\chi$ which is found to immediately lead to a CSL[70,71] in this range of parameters. This is consistent with perturbing the Dirac spin liquid with an explicit mass term $m\bar{\psi}\psi$, which is immediately generated on breaking time reversal and parity symmetry, leading to a chiral spin liquid. Outside this parameter range, the CSL is also obtained, but only on adding a finite value of $J_\chi$.

Given the relatively small values of $J_2$ involved, even the nearest-neighbor Heisenberg model should display aspects of spin liquid physics at intermediate scales which can be studied in future experiments and numerics. There are relatively few experimental candidates for the triangular lattice $S = 1/2$ materials that are undistorted. Two recently studied candidates are

$Ba_3Co$ $Sb_2O_9$ and $Ba_8CoNb_6O_{24}$. The latter compound fails to order even at the lowest temperatures measured $\sim J/25$[72], and is a promising spin liquid candidate. Although the former compound orders at low temperatures, its excitation spectrum[73] is hard to account for within spin wave theory. We note that at a qualitative level, the discrepancies from spin wave theory are connected to low-energy spectral weight at the M points, which are recovered within the QED$_3$ theory, where they arise from fermion bilinears. It has been pointed out that additional terms beyond the Heisenberg interaction may be present in these materials[74], which help stabilize a spin liquid state. The transition metal dichalcogenide 1T-TaS$_2$ has also been proposed as a quantum spin liquid where the spin degrees of freedom reside on clusters that form a triangular lattice[75]. The possibility of realizing the Dirac spin

liquid state on the triangular lattice should give additional impetus to physical realizations of $S = 1/2$ triangular lattice magnets for example, in ultracold atomic lattices and in twisted bilayers of transition metal dichalcogenides which remain to be experimentally realized[76].

In Fig. 4 we show the momenta and other spatial symmetry quantum numbers of the 15 fermion bilinears (adjoint masses with identical scaling dimension) and 1 singlet mass, and those of the six monopoles. These should help guide the search for Dirac spin liquids in X-ray (sensitive to the spin singlet excitations) and neutron (which probe both singlet and triplet excitations) scattering experiments.

In addition to a potentially stable spin liquid phase, which would represent a remarkable new state of matter, the DSL provides a unified picture to describe competing orders which have already been observed either in experiments or in numerical calculations, on different lattices. These range from the colinear Neel states on bipartite lattices to the 120° degree ordered states on the triangular and kagome lattices, and to spin singlet valence bond crystal states. It is hoped that such a unified picture of two-dimensional magnetism will deepen our understanding of some of the most interesting correlated electronic materials.

## Methods

**Numerical calculation of the monopole Berry phase**. We adopt the following approach to calculate the monopole quantum numbers. For concreteness, consider the $\pi$ flux theory on an $L \times L$ square lattice on the torus and uniformly spread the monopole flux of $2\pi q$, so that each plaquette contains an additional flux of $2\pi q/L^2$. We numerically verify that the energy spectrum has a finite size gap with exactly $4q$ zero modes. This is consistent with the theoretical expectation and justifies constructing monopole operators on a torus. We now need to decide which zero modes to fill before proceeding with the monopole quantum number calculation. Let us now specialize to the case of a single monopole, i.e. $q = 1$. Note, the following calculation will extract both the contribution from the zero modes, as well as the more elusive Dirac sea contribution (see Supplementary Note 2 for precise Berry phase definition). Since the latter is common to all flavors of monopoles, it suffices to consider just the monopole which corresponds to filling the whole Dirac sea plus the two spin-up zero modes, while leaving the two spin down zero modes empty. This monopole preserves the largest set of lattice symmetries and corresponds to the monopole operator $\Phi_4^\dagger - i\Phi_5^\dagger$. We then numerically calculate its lattice quantum numbers by evaluating $\langle \psi | G_R \cdot R | \psi \rangle$, where $G_R$ is a gauge transformation that keeps the Hamiltonian invariant after the lattice symmetry transformation $R$.

These quantum numbers are not all independent: they should satisfy the algebraic relations of the space group and time-reversal symmetry $\mathcal{T}$[28,36,38]. This greatly reduces the number of $U(1)_{top}$ phase factors to determine, and moreover it constrains the quantum numbers of monopoles to a discrete set of allowed solutions. For example, the momentum of the monopole in the kagome Dirac spin liquid must be 0 coming from the symmetry relation

$$T_1 C_6 T_2 = T_2 C_6. \tag{12}$$

Since monopole charge does not change under rotation or translation in this case, we can assign the Berry phase $\theta_C$, $\theta_{1/2}$ with $C_6$, $T_{1/2}$, respectively. Furthermore, from the fermion PSG which control fermion zero mode transformation (Supplementary Note 1), one knows the spin triplet monopoles $\mathcal{S}_i$'s stay invariant up to Berry phases under these symmetries and hence transform as

$$C_6, T_{1/2} : \mathcal{S}_i \rightarrow e^{i\theta_{C/1/2}} \mathcal{S}_i \tag{13}$$

combining with Eq. (12) immediately yields $\theta_1 = 0$. Similarly one gets vanishing Berry phase for all translations. An intuitive way to see this is that the $\mathcal{S}_i$'s momentum should stay invariant under $C_6$ and the only $C_6$ invariant point in Kagome Brillouin zone is the $\Gamma$ point. This is not the case on triangular lattice where Berry phase for translation is $\pm 2\pi/3$. The key point is that on the triangular lattice, the PSG for $C_6$ involves charge conjugation to compensate for the exchange of triangles with $0, \pi$ gauge flux, and hence monopoles go to anti-monopoles. The nontrivial phase factor of the kagome Dirac spin liquid is associated with the $C_6$ rotation about the plaquette center[28] and we use the aforementioned numerical scheme to directly read off this rotation quantum number. We list the possible Berry phases compatible with algebraic relations of symmetry group on four lattices in Supplementary Note 1 and below is the numerically found Berry phase which indeed falls among one of the possible choices constrained by algebraic relations.

On the triangular lattice, the lattice momentum of the monopole $\Phi_4^\dagger$ is constrained, by point group symmetries, to be one of $0, \pm 2\pi/3$. We determine its precise value here by evaluating $\langle \psi | G_{T_{1,2}} \cdot T_{1,2} | \psi \rangle$, i.e. combining translation $T_{1,2}$

with the appropriate gauge transformation $G_{T_{1,2}}$ to leave the mean field ansatz invariant. Again, we consider filling the Dirac sea and the two spin up zero modes as our ground state $|\psi\rangle$. Unfortunately, one cannot resort to reading off the relevant quantum numbers since the Hamiltonian on a torus with the added flux is not translation invariant. To see this, note that the phase factors of Wilson loops $e^{\oint_C a \cdot dl}$, along two adjacent columns (rows) will always differ by $2\pi/L$ (there are $L$ unit cells between them). Such translation symmetry breaking will be invisible in the thermodynamic limit, since the phase difference of adjacent Wilson loops $2\pi/L \rightarrow 0$ as $L$ increases. We expect that translation symmetry to be recovered for large enough system sizes. Indeed, this is what we observe in Fig. 4, which shows our numerical results for $\langle \psi | G_{T_1} \cdot T_1 | \psi \rangle = \rho e^{ik_1}$. For a large $L$, the momentum $k_1$ is perfectly quantized to $2\pi/3$. Moreover, the amplitude $\rho$ approaches unity, indicating the restoration of translation symmetry. As a comparison, we also calculate the monopole's momentum on the square lattice, and we get $k_1 = \pi$. We choose a Landau gauge on square lattice, i.e.

$$A_{i,i+\hat{x}} = By_i,$$
$$A_{i,i+\hat{y}} = \begin{cases} 0 & y_i \neq L_y - 1 \\ BL_y x_i & y_i = L_y - 1 \end{cases} \tag{14}$$

where $i$ labels square lattice site with coordinate $(x_i, y_i)$ on a system of size $L_x \times L_y$ and $B = 2\pi/(L_x L_y)$ is the uniform field strengthl. Upon translation along $x$ direction by one unit $T_1$, only $A_{i,i+\hat{y}}$ at $y = L_y - 1$ changes and can not be compensated by a gauge transform. The change $2\pi/L_x$, however goes to 0 in the thermodynamic limit. Hence in this case one does not need to do the gauge transform in numerics $G_{T_1}$ and can directly compute the wavefunction overlap. Similar gauge choice on triangular lattice also gives a trivial $G_{T_1}$. Therefore, the momentum of the monopole $\Phi_4^\dagger$ on the $\pi$ flux square lattice is $(\pi, \pi)$, on the triangular lattice it is quantized to $(-2\pi/3, 2\pi/3)$ and $C_6$ equivalents (the gauge on triangular lattice is similar). We also numerically verify that under translations on honeycomb and kagome lattices, monopoles has zero Berry phase dictated by algebraic relations. For triangular lattice, since reflection involves charge conjugation, monopole charge does not change and numerically we find the Berry phase under reflection to be 0.

We note that our result for the square lattice is consistent with an earlier calculation[38] which used a cylinder geometry to select from the discrete set of possibilities fallowed by crystal symmetries. The cylinder geometry is particularly convenient since translation symmetry in one direction can be preserved even in the presence of flux. Unfortunately, we caution that for other lattices, the cylinder geometry gives an incorrect answer (i.e. $\pi$ even for the triangular lattice where the only consistent momenta are $0, \pm 2\pi/3$). We believe this problem potentially arises from the presence of edge states, which can be circumvented by adopting the torus geometry as we have done here.

Finally, we remark that our numerical method cannot determine the precise transformation of the time-reversal symmetry $\mathcal{T}$ and reflections. This, however, can be calculated analytically as we will describe elsewhere[49], which also confirm the results here for other symmetries. It turns out that for all the Dirac spin liquid states we consider, the spin-singlet monopoles are even under time-reversal $\Phi_{1,2,3}^\dagger \rightarrow \Phi_{1,2,3}$, while the spin triplet monopoles are odd under the time-reversal $\Phi_{4,5,6}^\dagger \rightarrow -\Phi_{4,5,6}$. It contrast, ref. [28] conjectured that all monopoles in the kagome Dirac spin liquid are odd under $\mathcal{T}$.

## Data availability
The data that support the findings of this study are available from the corresponding authors upon request.

## Code availability
The codes to compute monopole quantum numbers are available from the authors upon request.

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

## Acknowledgements

We gratefully acknowledge helpful discussions with Cristian Batista, Chao-Ming Jian, Max Metlitski, Adrian H.C. Po, Ying Ran, Subir Sachdev, Cenke Xu, Yi-Zhuang You, and Liujun Zou. X.-Y.S. acknowledges hospitality of Kavli Institute of Theoretical Physics (NSF PHY-1748958). A.V. was supported by a Simons Investigator award. C.W. was supported by the Harvard Society of Fellows. Y.-C.H. was supported by the Gordon and Betty Moore Foundation under the EPiQS initiative, GBMF4306, at Harvard University. Research at Perimeter Institute (Y.-C.H. and C.W.) is supported by the Government of Canada through the Department of Innovation, Science and Economic Development Canada and by the Province of Ontario through the Ministry of Research, Innovation and Science.

## Author contributions

X.-Y.S., C.W., A.V. and Y.-C.H. participated in the theoretical construction and manuscript preparation. Y.-C.H. performed the numerical calculations of monopole quantum numbers.

## Additional information

**Competing interests:** The authors declare no competing interests.

