## [Peer Review File · Nature Communications]

Reviewers' comments:

Reviewer #1 (Remarks to the Author):

In this manuscript the authors provide a compelling analysis of symmetry breaking transitions out of the Dirac spin liquid (DSL) state in two dimensional quantum magnets. They argue that the DSL can be viewed as a kind of parent state from which various competing ordered states can be understood, thereby providing a unified description of different symmetry broken states. Two key ingredients are provided to achieve this goal. First, the authors provide a complete symmetry analysis of the monopole excitations for square, honeycomb, triangular and kagome lattices, including Berry phase contributions from the Dirac sea. A second key insight is the mechanism which gives rise to confinement and thus to symmetry breaking orders. In particular, the authors argue that this happens in two steps: first a fermion mass term is spontaneously generated, for which there are 16 different possibilities due to the SU(4) flavor symmetry of the DSL. Secondly, a corresponding monopole is proliferated which gives rise to confinement. The authors argue that this monopole is uniquely selected because the fermion mass term splits the degeneracy of fermion zero modes associated with the monopoles. Finally, using the symmetry properties of the condensed monopole the authors infer the corresponding order: while condensing a spin-singlet monopole gives rise to a specific valence bond solid order, condensing a spin-1 monopole leads to specific types of magnetic order.

The manuscript is in parts rather concise (despite its length), but it presents the results in sufficient detail. In a related preprint (Ref. 47) the authors present a detailed study of symmetry properties of monopoles as well, but from the perspective of spinon band topology, which is quite different from this work. Also, the main focus of this manuscript is on symmetry breaking orders, which is not touched upon in Ref. 47. The comprehensive analysis of symmetry breaking orders due to monopole proliferation presented here had not been achieved before. It is a significant advancement in the field of quantum magnetism, which is definitely of great interest in the community, and in my opinion deserves to be published in Nature Communications.

I have a few questions and remarks which the authors might want to consider before publication:

1.) It would be very interesting if the authors have any comment or insight on why the construction presented here can only give rise to very specific types of order (e.g. 120 degree magnetic order). For example, various other types of magnetic order have been discussed in the literature for the kagome lattice, such as $q = \sqrt{3} \times \sqrt{3}$ or the chiral cuboc orders, which are not touched upon in this work. Does the DSL approach presented here allow to draw conclusions about the stability of the $q=0$ versus the other states in particular circumstances? Also, would it be possible to get chiral (non-coplanar) magnetically ordered states by generating both, a singlet and a chiral mass term for the fermions?

2.) There are some subtleties with the computation of the monopole Berry phases, which I think should be addressed by the authors. First, the authors uniformly spread the monopole flux over the whole lattice. While it is reasonable to spread the flux over distances larger than the lattice spacing, spreading it over the entire lattice seems to be a stretch, in particular since the monopole is supposed to be a local excitation. Second, and more importantly, the authors compute the Berry phase within the mean-field model where the constraint of no double occupancy is not accounted for when they fill the Dirac sea. Since a key property is the monopole Berry phase which it picks up when encircling a lattice site, this constraint seems to be rather crucial. It would be helpful if the authors could comment on these points.

Reviewer #2 (Remarks to the Author):

The submitted manuscript "Unifying Description of Competing Orders in Two Dimensional Quantum Magnets" is a highly relevant theoretical study of the properties (quantum numbers) of monopoles in the QED3 description of so called Dirac Spin Liquids (DSL). These DSL are prominent candidates to describe the low energy properties of many (frustrated) quantum magnets. The present work is a milestone in that it clearly identifies the quantum numbers of the presumably most relevant monopole operators in the theory which will dictate the nature of the quantum phases in the vicinity of the DSL phase/point. This was a long-standing open question in the field. The paper is also clearly separated from a second arXiv preprint by the same authors, where they mostly describe a technology to obtain these quantum numbers, while in the present manuscript the technology is not central, but the implications of these results are outlined. I strongly support the acceptance of this work for publication in Nature Communications. The only criticism I have right now is that the English and the grammar could be improved at places.

Reviewer #3 (Remarks to the Author):

In this paper the authors expand on the idea that the Dirac spin liquid is the mother of many competing orders (a point of view previously forcefully advocated in the manuscript's ref 28), by a detailed analysis of the manner in which lattice symmetries are realized in an abelian gauge theory description of the state, in particular on the monopole operators.

For a spin system on any lattice, we can formulate a parton construction leading to a Dirac spin liquid -- U(1) gauge theory with N_f massless Dirac fermions -- as a candidate state.

It has been known for some time that in the parton construction, the structure of the lattice is remembered by the lattice-symmetry quantum numbers of the monopole operators; for example, this plays a crucial role in the question of the stability of the deconfined critical point between Neel and VBS phases.

But it is fair to say that a systematic analysis, in particular for fermionic partons, and with a unified point of view for all lattices, has been lacking (one was promised long ago as forthcoming work as ref 35 of the manuscript's ref 28). This paper (with its companion) accomplishes this important task.

In addition to allowing to address questions of stability of the state and the nature of nearby ordered phases, these quantum numbers clarify the experimental signatures of such a Dirac spin liquid state (at which reciprocal lattice vectors should various excitations appear).

The paper gives an interesting analysis of the implications for the fate of the Dirac spin liquid point under generic symmetric perturbations, for various lattices, finding that the monopoles which want to proliferate have appropriate quantum numbers to produce familiar complex ordered phases on those lattices. In particular, the authors develop a lovely connection between which fermion bilinear condensate lifts the fermions and which monopoles proliferate. They also make contact with the deconfined critical point between various such orders and its enlarged symmetry group.

This paper is distinguished from its companion in that it uses a numerical method (refining earlier work of Alicea) to identify the monopole charges under various symmetries, while its companion paper gives an argument (based on band topology) for their charges under symmetries to which the numerical method cannot be applied. Taken together this work is an important advance.

I think it should be published.

I have a few minor comments and questions:

-- On page 4 the authors say "Clearly spin-rotation can only be embedded as an $SO(3)$ subgroup of the $SO(6)$ flavor group, meaning that three of the six monopoles form a spin-1 vector, and the other three are spin singlets." It is not so obvious to me, a priori: without physics input (or the explicit representation in equation (6)), there are other ways of embedding $SO(3)$ into $SO(6)$, for example as $SO(3) \subset SU(2) \subset SU(4) = SO(6)$, in which case the monopoles would end up in a spinor (i.e. 1/2-integer spin) representation.

But perhaps this is immediately ruled out by a related question/confusion I have: why do the authors say that the spin rotation group is $SO(3)$ and not $SU(2)$? We are talking about spin 1/2 systems, no? So the spin rotation group is $SU(2)$ and half-integer spin representations are linear (not projective) representations. This seems to be contradicted by the statement (also on page 4):

"An important observation here is that since monopoles are local operators, they transform as linear representations of the symmetry group, in contrast to gauge charged fermions that transform under a projective symmetry group. Thus, for example, the monopoles transform as integer spin representations, unlike the spinons which carry spin one half."

But I believe this is just a confusion of language: even acting on a half-integer spin Hilbert space, the spin operators transform as integer spin representations -- there are in fact no gauge-invariant local operators which transform with half-integer spin. So I conclude that acting on gauge-invariant local operators, the spin group is indeed $SO(3)$, and believe this is what the authors meant. If the authors agree with my understanding, it would be helpful to make this clearer in the text.

-- The authors propose an interesting and appealing analogy between the algebraic spin liquid in $2+1d$ and the Luttinger liquid in $1+1d$. In particular the authors claim that the Luttinger liquid can be reformulated in terms of QED_{1+1} . This is not so well known and a brief summary of how this works (perhaps in an appendix) would be welcome.

It would strengthen the analogy the authors are suggesting.

-- The two-step procedure to reach ordered phases advocated at the beginning of section IV.B (first fermion bilinear condensation, then monopole proliferation) seems to be not inevitable. In fact, section IV.B.1, which discusses the case with invariant monopoles, actually reverses the two steps.

-- There are minor typos and infelicities of language which should be repaired, such as
- the extra period in the sentence

"For example, the momentum of the monopole in the kagom'e Dirac spin liquid must be 0. coming from the symmetry relation..."

- the word "vacuaa".

-- John McGreevy

Dear editors and referees,

We thank your thoughtful feedback, hereby submit the revised manuscript and below give a summary of revisions and address referees' comments.

Changes made to the manuscript:

1. Add reference [12] on the relations of Luttinger liquid and QED2.
2. Modify the sentence in sec II above eq (6), "Clearly spin-rotation,..." to emphasize that spin rotation acts as an $SO(3)$ group on physical gauge-invariant operators.
3. Refine the discussion on symmetry-allowed monopole term on Kagome & triangular lattices eq (7), (8) in terms of higher order monopoles with different Lorentz spin reps and an appendix IV. Correct the scaling dimension of the first excited monopole operators in (8) due to Lorentz spin considerations.
4. Add a sentence in the middle of first paragraph of sec IIIB "This particular ordering... our purpose" to clarify mass condensation + monopole proliferation is not the sole mechanism driving instability.
5. Elaborate the gauge used on square lattice in numerical calculations (eq (14) and immediate explanations).
6. Change the symbols in figure 4 on experimental signatures of DSL.
7. Incorporate footnotes into main text and correct typos.
8. Add code availability statements before Acknowledgements.

Response to specific referee comments:

Reviewers' comments:

Reviewer #1 (Remarks to the Author):

I have a few questions and remarks which the authors might want to consider before publication:

1.) It would be very interesting if the authors have any comment or insight on why the construction presented here can only give rise to very specific types of order (e.g. 120 degree magnetic order). For example, various other types of magnetic order have been discussed in the literature for the kagome lattice, such as $q = \sqrt{3} \times \sqrt{3}$ or the chiral cuboc orders, which are not touched upon in this work. Does the DSL approach presented here allow to draw

conclusions about the stability of the $q=0$ versus the other states in particular circumstances? Also, would it be possible to get chiral (non-coplanar) magnetically ordered states by generating both, a singlet and a chiral mass term for the fermions?

Indeed, the monopole proliferation in DSL does not lead to the other spin orders mentioned by Referee A and we believe this means these orders are not proximate to DSL. But other orders may develop into spin liquid phase other than DSL, so we'd prefer not to give definite answer to stability of these orders.

A singlet mass will generate a Chern-Simons term for 'a' field and prevent monopoles from proliferating. From analysis of the other 15 masses + monopole we didn't find non-coplanar magnetic order.

2.) There are some subtleties with the computation of the monopole Berry phases, which I think should be addressed by the authors. First, the authors uniformly spread the monopole flux over the whole lattice. While it is reasonable to spread the flux over distances larger than the lattice spacing, spreading it over the entire lattice seems to be a stretch, in particular since the monopole is supposed to be a local excitation. Second, and more importantly, the authors compute the Berry phase within the mean-field model where the constraint of no double occupancy is not accounted for when they fill the Dirac sea. Since a key property is the monopole Berry phase which it picks up when encircling a lattice site, this constraint seems to be rather crucial. It would be helpful if the authors could comment on these points.

Since the monopole momenta are quantized and Berry phase is most readily computable when the system has translation invariance, it is reasonable to spread the flux evenly while preserving the Berry phase. We believe this is a useful approximation to a well spread out flux.

We are interested in the kinematical properties of monopoles which stay invariant through adiabatic deformation of the system (e.g., from one parton per site to the mean-field configuration in numerics). Actually a parallel paper (arxiv:1811.11182) exploits this fact to deform the parton band to a Wannier insulator (centers don't necessarily coincide with sites) to analytically determine Berry phase. Of course we assume the effect of gauge fluctuations (the implementation of projection at the field theory level) is not so severe as to completely change the mean field result, especially when one is just choosing from a set of discrete choices. Post facto, the proximity of well known magnetic orders such as the 120 degree state on triangular lattice provides some support for this assumption.

Reviewer #3 (Remarks to the Author):

-- On page 4 the authors say "Clearly spin-rotation can only be embedded as an $SO(3)$ subgroup of the $SO(6)$ flavor group, meaning that three of the six monopoles form a spin-1 vector, and the other three are spin singlets." It is not so obvious to me, a priori: without physics input (or the explicit representation in equation (6)), there are other ways of embedding $SO(3)$ into $SO(6)$, for example as $SO(3) \subset SU(2) \subset SU(4) = SO(6)$, in which case the

monopoles would end up in a spinor (i.e. 1/2-integer spin) representation.

But perhaps this is immediately ruled out by a related question/confusion I have: why do the authors say that the spin rotation group is $SO(3)$ and not $SU(2)$? We are talking about spin 1/2 systems, no? So the spin rotation group is $SU(2)$ and half-integer spin representations are linear (not projective) representations. This seems to be contradicted by the statement (also on page 4):

"An important observation here is that since monopoles are local operators, they transform as linear representations of the symmetry group, in contrast to gauge charged fermions that transform under a projective symmetry group. Thus, for example, the monopoles transform as integer spin representations, unlike the spinons which carry spin one half."

But I believe this is just a confusion of language: even acting on a half-integer spin Hilbert space, the spin $\{it\}$ operators transform as integer spin representations -- there are in fact no gauge-invariant local operators which transform with half-integer spin. So I conclude that $\{it\}$ acting on gauge-invariant local operators, the spin group is indeed $SO(3)$, and believe this is what the authors meant. If the authors agree with my understanding, it would be helpful to make this clearer in the text.

We agree. Please see change #2.

-- The authors propose an interesting and appealing analogy between the algebraic spin liquid in $2+1d$ and the Luttinger liquid in $1+1d$. In particular the authors claim that the Luttinger liquid can be reformulated in terms of QED_{1+1} . This is not so well known and a brief summary of how this works (perhaps in an appendix) would be welcome. It would strengthen the analogy the authors are suggesting.

We've added reference [12].

-- The two-step procedure to reach ordered phases advocated at the beginning of section IV.B (first fermion bilinear condensation, then monopole proliferation) seems to be not inevitable. In fact, section IV.B.1, which discusses the case with invariant monopoles, actually reverses the two steps.

See change #4.

-- There are minor typos and infelicities of language which should be repaired, such as
- the extra period in the sentence
"For example, the momentum of the monopole in the kagom'e Dirac spin liquid must be 0. coming from the symmetry relation..."
- the word "vacuaa".

See change #7.

REVIEWERS' COMMENTS:

Reviewer #1 (Remarks to the Author):

The authors have responded to all of the referee's questions and comments and updated their manuscript accordingly. All open issues have been addressed and I strongly recommend to publish this manuscript in its current form.

Reviewer #3 (Remarks to the Author):

Looks good.

Change #2 introduced a typo "afnd" instead of "and".